# Aphrodisiac Performance of Bioactive Compounds from *Mimosa pudica* Linn.: In Silico Molecular Docking and Dynamics Simulation Approach

**DOI:** 10.3390/molecules27123799

**Published:** 2022-06-13

**Authors:** Chandrasekar Palanichamy, Parasuraman Pavadai, Theivendren Panneerselvam, Sankarganesh Arunachalam, Ewa Babkiewicz, Sureshbabu Ram Kumar Pandian, Kabilan Shanmugampillai Jeyarajaguru, Damodar Nayak Ammunje, Suthendran Kannan, Jaikanth Chandrasekaran, Krishnan Sundar, Piotr Maszczyk, Selvaraj Kunjiappan

**Affiliations:** 1Department of Biotechnology, Kalasalingam Academy of Research and Education, Krishnankoil 626126, India; p.chandrasekar@klu.ac.in (C.P.); sankarganesh@gmail.com (S.A.); srkpandian@klu.ac.in (S.R.K.P.); kabilan@klu.ac.in (K.S.J.); 2Department of Pharmaceutical Chemistry, Faculty of Pharmacy, M.S. Ramaiah University of Applied Sciences, M S R Nagar, Bengaluru 560054, India; pvpram@gmail.com; 3Department of Pharmaceutical Chemistry, Swamy Vivekanandha College of Pharmacy, Tiruchengodu 637205, India; tpsphc@gmail.com; 4Department of Hydrobiology, Faculty of Biology, University of Warsaw, 02-089 Warsaw, Poland; ewa.babkiewicz@wp.pl; 5Department of Pharmacology, Faculty of Pharmacy, M.S. Ramaiah University of Applied Sciences, M S R Nagar, Bengaluru 560054, India; superdamu@gmail.com; 6Department of Information Technology, Kalasalingam Academy of Research and Education, Krishnankoil 626126, India; k.suthendran@klu.ac.in; 7Department of Pharmacology, School of Pharmacy & Technology Management, SVKM’S NMIMS University, Secunderabad 500017, India; jaikanthjai@gmail.com

**Keywords:** bioactive molecules, *Mimosa pudica* Linn., molecular docking, molecular dynamics simulation, penile erection

## Abstract

Plants and their derived molecules have been traditionally used to manage numerous pathological complications, including male erectile dysfunction (ED). *Mimosa pudica* Linn. commonly referred to as the touch-me-not plant, and its extract are important sources of new lead molecules in drug discovery research. The main goal of this study was to predict highly effective molecules from *M. pudica* Linn. for reaching and maintaining penile erection before and during sexual intercourse through in silico molecular docking and dynamics simulation tools. A total of 28 bioactive molecules were identified from this target plant through public repositories, and their chemical structures were drawn using Chemsketch software. Graph theoretical network principles were applied to identify the ideal target (phosphodiesterase type 5) and rebuild the network to visualize the responsible signaling genes, proteins, and enzymes. The 28 identified bioactive molecules were docked against the phosphodiesterase type 5 (PDE5) enzyme and compared with the standard PDE5 inhibitor (sildenafil). Pharmacokinetics (ADME), toxicity, and several physicochemical properties of bioactive molecules were assessed to confirm their drug-likeness property. Molecular dynamics (MD) simulation modeling was performed to investigate the stability of PDE5–ligand complexes. Four bioactive molecules (Bufadienolide (−12.30 kcal mol^−1^), Stigmasterol (−11.40 kcal mol^−1^), Isovitexin (−11.20 kcal mol^−1^), and Apigetrin (−11.20 kcal mol^−1^)) showed the top binding affinities with the PDE5 enzyme, much more powerful than the standard PDE5 inhibitor (−9.80 kcal mol^−1^). The four top binding bioactive molecules were further validated for a stable binding affinity with the PDE5 enzyme and conformation during the MD simulation period as compared to the apoprotein and standard PDE5 inhibitor complexes. Further, the four top binding bioactive molecules demonstrated significant drug-likeness characteristics with lower toxicity profiles. According to the findings, the four top binding molecules may be used as potent and safe PDE5 inhibitors and could potentially be used in the treatment of ED.

## 1. Introduction

Male sexual inadequacy, including erectile dysfunction (ED) or impotence, organ disorder, pain during intercourse, loss of libido, premature ejaculation, hypogonadism, lack of sexual desire, etc., is a serious health concern facing both young and old men worldwide [1]. Sexual satisfaction and relationships have long been significant components for developing and maintaining a social and biological relationship in human life [2]. Sexual dysfunction (SD) has a significant negative impact on an individual’s health, quality of life, and life expectancy [3]. The occurrence of SD increases with age, but sexual problems in men over the age of 60 and, especially, over the age of 70 is often lower, explaining the decreasing incidence of clinically relevant ED [4]. It has been projected that ED will affect one-third of a billion men worldwide by 2025 [5]. ED is the persistent inability to reach and retain penile erection, which results in unsatisfactory sexual intercourse [6]. An exact balance of psychological, hormonal, neurological, vascular, and cavernosal factors are required for normal erectile function. Alterations in any one or combinations of these factors may lead to ED. The root cause of the alterations of these factors might be injury, neurological disorder, hormonal imbalance, cardiovascular diseases, anxiety, heavy workload, and stress associated with a sedentary lifestyle [7].

Many previous investigations have demonstrated that oxidative stress and reactive oxygen species (ROS) play a significant role in the pathophysiology of ED [8,9,10,11]. Oxidative stress occurs due to the imbalance between free radicals (prooxidants, hydroxyl radical (OH *), and nitric oxide (NO *)) and the ability of antioxidants to scavenge excessive reactive oxygen species [10,11,12,13]. Though the role of ROS and oxidative stress has been proven in the pathophysiological mechanisms of male and female infertility [14], its influence in ED has not been investigated systematically. However, preliminary studies have shown a significant connection between the production of ROS and ED, particularly in diabetic animal models [8].

Recently, various treatment strategies are being used to treat ED based on the severity and underlying health conditions. Commonly used treatment strategies are oral phosphodiesterase type 5 (PDE5) inhibitors such as sildenafil (Viagra), tadalafil (Cialis), vardenafil (Levitra), and avanafil (Stendra); synthetic prostaglandin E1 (alprostadil); phentolamine; papaverine; exercise; moderate to vigorous aerobic activity; herbal Viagra; vacuum erection devices; surgical penile implants; penis pumps; and hormonal treatments—specifically with testosterone, which increases firmness, maintaining of a penile erection, frequency of orgasm, and level of desire [15,16,17]. However, such treatment options have been linked to undesirable consequences and even death. The consequences include post-surgical infections, intense pain from mechanical devices, and, in addition, most of these procedures are expensive [18]. Moreover, widely used PDE5 inhibitors also cause serious adverse effects, such as headache, myalgia, facial flushing, heartburn, nasal congestion, and vision-related issues, because the PDE5 gene is widely presented in the human body [19]. Moreover, pathophysiological conditions of endothelial dysfunction affect the impaired release of NO, which leads to other complications, including cardiovascular diseases, diabetes mellitus, etc. [20]. Hence, researchers and clinicians are in search of safe and cost-effective medicines for the treatment of ED. Since ancient times, people around the world have been using plants as medicine to treat numerous complications, including SD [21]. Since plants contain a plethora of bioactive components, they are responsible for alleviating various pathological complications. Herbs and their formulations have been used as aphrodisiacs throughout the history to enhance the libido, maintain penile erections, and improve sexual desire and endurance, as well as energy [22].

*Mimosa pudica* Linn. (Mimosaceae) is a creeping annual or biannual sub-woody plant native to the tropics of Central and South America, which is also found in Australia and India. It is commonly referred to as the ‘touch-me-not’ plant or ‘sensitive plant’ or ‘humble plant’ for its fascinating leaf movement when touched [23]. The extract of *M. pudica* Linn. has been used traditionally in the treatment of various human afflictions, including aging and urogenital diseases. It is also used as an antidiabetic [24], acetylcholinesterase inhibitor [25], antifertility [26], antivenom [27], antidepressant [28], anticonvulsant [29], aphrodisiac [30], anti-ophidian [31], etc. Several studies have demonstrated that the phytochemical profile of the *M. pudica* Linn. extract has β-carotene, dl-norepinephrine, gallic acid, mimosine, mimopudine, turgorin, amino acids, flavonoids, triterpenes, etc., which are being used for the treatment of several complications [32]. The aim of this study was to predict highly effective molecules from *M. pudica* Linn. for reaching and maintaining a penile erection before and during sexual intercourse through in silico molecular docking and dynamics simulation tools against PDE5. To achieve this goal, the signaling network of PDE5 was examined through the analysis of a graph theoretical network by which it may be able to comprehensively study the topology and functions of the selected network. This network analysis can also help in the selection of an ideal drug target that can specifically suppress or stimulate the receptor [33]. Further, the pharmacokinetic and physicochemical properties of the screened bioactive molecules were also studied. The molecular stability and binding interactions of the selected bioactive molecules with the PDE5 protein were also evaluated using molecular dynamics simulation methods.

## 2. Experimental Section

### 2.1. Bioactive Compounds Identification

The Indian Medicinal Plants, Phytochemistry and Therapeutics (IMPPAT) database [34], previously published articles, and the PubChem database were utilized to find 28 bioactive molecules from *M. pudica* Linn., along with a standard drug, sildenafil. For in silico molecular docking studies, the three-dimensional structures (Structure Data File format) of selected bioactive components were modeled using Chemsketch and energy optimized with MMFF94 (Merck molecular force field 94).

### 2.2. Graph Theoretical Network Investigation

The network was retrieved from the Kyoto Encyclopedia of Genes and Genomes (KEGG) database, and the graph theoretical network analysis using the selected signaling pathway was reconstructed by Cytoscape version 3.7.1 software [35,36]. The influence of various genes, proteins, enzymes, and their roles in the PDE5 signaling pathway (hsa04022) in *Homo sapiens* was visualized.

### 2.3. Protein Preparation

The RCSB PDB (Research Collaboratory for Structural Bioinformatics, Protein Data Bank) website provided the coordinate file and X-ray crystal structure of the human phosphodiesterase type 5 (PDE5) protein domain in complex with sildenafil (PDB entry ID: 1udt, resolution 2.30 Å) [37]. The presence of improper bonds and side chain anomalies were corrected and missing residues were introduced in the retrieved protein using the Swiss-PDB Viewer v4.1.0. The file was assigned the name target.pdb and was stored for future exploration. In addition, we used BIOVIA Discovery Studio Visualizer version 4.0 software (Accelrys Software Inc., San Diego, CA, USA) to define the protein structure and amino acid position from active sites, which was then used for the molecular docking experiments [38].

### 2.4. Active Binding Site Prediction

Predicting ligand binding sites from particular sites of a protein structure has numerous implications in protein function elucidation and structure-based drug development [39]. This binding site prediction helps the bioactive molecules to bind and establish sufficient interaction with the target protein, as well as to create a robust ligand–target site interaction to provide the optimal and favorable catalytic effects. All possible active binding sites of the targeted bioactive molecules were determined using the PrankWeb (https://prankweb.cz/) online tool for further investigation. Once the active binding site of the target protein was predicted, a receptor grid box was generated using the PyRx program.

### 2.5. Molecular Docking

We used the PyRx 0.8 tool in the AutoDock Vina program to explore the molecular docking of selected bioactive molecules against the PDE5 protein. The ligands were the bioactive molecules identified from *M. pudica* Linn., and the receptor was the PDE5 protein. The protein (receptor) and bioactive molecule (ligands) files were saved in the “.pdbqt” format to evaluate the molecular binding affinities (kcal mol^−1^) where the grid box with a 10.0 Å radius throughout the active binding site predicted region was chosen. The AutoDock Vina program measured the binding energy affinities of up to 10 distinct docking sites for each ligand. All complex binding affinity energies were calculated on the basis of ligand conformation at the active binding site, with the RMSD (root mean square deviation) between the original and then the structures taken into account. The number of hydrogen bonds and noncovalent interactions for each complex was calculated using Discovery Studio Visualizer, which produced details, compound images, and interaction images (2D and 3D) [40].

### 2.6. Pharmacokinetics and Physicochemical Properties Prediction

The drug-likeness, pharmacokinetics, and toxicity properties of the potential lead molecules are essential in terms of reducing toxicity and increasing the bioavailability in the pharmaceutical industry [41]. Here, we used web-based Swiss-ADME and pkCSM-pharmacokinetic tools to determine the pharmacokinetics (absorption, distribution, metabolism, and excretion); safety; and physicochemical properties of the selected bioactive molecules from *M. pudica* Linn.

### 2.7. Molecular Dynamics (MD) Simulation Studies

MD simulation is a helpful technique for the current drug discovery processes in understanding molecular structure-to-function relationships [42]. In the present study, MD simulation methodology was used to evaluate the binding stability, conformation, and interaction modes between the chosen bioactive molecules (ligands) and target protein PDE5. MD simulation investigations of the receptor–ligand complexes were carried out for 100 ns using GROMACS (Groningen Machine for Chemical Simulations) software version 2019.2 [43]. GROMACS software has excellent throughput and a very parallel open-source molecular simulation toolkit. The compound with the best binding-affinity to PDE5 complex files were chosen for MD simulation investigations. Selected bioactive molecule topology was retrieved from the PRODRG server, and the complex systems were prepared using the previously described method. In the MD simulation, the initial vacuum was lowered using the steepest descent approach for 5000 steps [44]. The protein was solvated with a simple point-charge (SPC) three-point water model, which has excellent computational efficiency and is commonly preferred for MD simulation studies [45]. A cubic periodic box of 1.2 nm was constructed, and a minimum distance of 6.0 Å was maintained between the protein and border of the simulation box to completely solvate the protein molecule and rotate freely, where the solvated system comprised a charged protein with a net charge of +8e [46]. Later, enough Na^+^ and Cl^−^ counter ions were introduced to keep the salt concentration in the complex system at 0.15 M. From the NPT (Isothermal-Isobaric, constant number of particles, pressure, and temperature) equilibration, each complex was assigned a simulation time of 50 ns for the final run. Using the internet server “WebGRO” for macromolecular simulations (https://simlab.uams.edu/), the root mean square deviation (RMSD), root mean square fluctuation (RMSF), radiation of gyration (RGY), and SASA (solvent accessible surface area) of the trajectory analysis were executed in the GROMACS simulation program [47].

### 2.8. Molecular Mechanics Poisson–Boltzmann Surface Area (MM/PBSA) Calculation

The MM/PBSA approach was used to calculate the protein–ligand binding free energy of each complex. The binding free energy (ΔG) was measured utilizing the GROMACS software tool g_mmpbsa [48].

### 2.9. Density Functional Theory (DFT)

The electronic effect of the drug-like molecule played an essential role in pharmacological activity. DFT is an effective theory for measuring the electronic states of atoms, molecules, and solids in terms of the three-dimensional electronic density system [49,50,51,52]. The main goal of DFT is to provide a quantitative understanding of material properties using the fundamental laws of quantum mechanics. In this study, the top binding scored bioactive compounds were obtained from molecular docking, and computational calculations were performed using the Gaussian 03W program and the GaussView molecular visualization tools. The DFT/Becke-3–Lee–Yang–Parr (B3LYP) technique was used to optimize the molecular structures of the selected bioactive molecule using a 6-311G (d,p) basis set. The frontier molecular orbital energies of the selected bioactive compounds (lowest unoccupied molecular orbital (ELUMO), highest occupied molecular orbital (EHOMO), and their energy gap (Eg)) were calculated using the optimized structures. The molecular orbital energy diagrams obtained for the selected bioactive compounds were displayed using GaussView, a molecular visualization tool.

## 3. Result

### 3.1. Selection of Bioactive Compounds and Preparation

From the IMPPAT database and published articles, three-dimensional structures of 28 bioactive molecules from the target plant *M. pudica* Linn., along with one standard drug, the PDE5 inhibitor, were selected and optimized. The optimized structures were used in in silico molecular docking investigations against the PDE5 enzyme, and the results are presented in Appendix A.

### 3.2. Graph Theoretical Network Analysis

The selected PDE5 signaling pathway was reconstructed using Cytoscape software as a graph by proteins or genes (nodes) and interactions (edges) and is shown in Figure 1 and Table 1. The reconstructed signaling network of PDE5 possessed 86 edges and one node, and the significance of the proteins was identified by calculating the centrality parameters. The calculated centrality measure values of betweenness (4355.81), closeness (0.0045), degree (20), eccentricity (0.1111), eigenvector (0.6646), radiality (10.963), and stress (6142) exposed the threshold values of all measures, as well as significant node in the network. On the basis of centrality calculations and threshold values, the protein PDE5 was identified as a significant therapeutic target for ED.

### 3.3. Active Binding Site Identification

As shown in Figure 2, the PDE5 protein (PDB ID: 1udt) evaluation results using the PrankWeb tool revealed that there are three potential binding sites with the ligands. Three binding pockets were predicted with different colors (blue, red, and green). The first binding pocket (blue color) is the highest score of all three pockets with a pocket score of 67.90, 36 amino acids, and a probability score of 0.990. The second binding pocket (red color) consists of a pocket score of 1.23, seven amino acids, and a probability score of 0.012. Finally, the third binding pocket (green color) had the smallest pocket score of 1.15, nine amino acids, and a probability score of 0.010. Molecular screening of the selected bioactive molecules was examined using the same investigated binding pockets of the PDE5 protein structure. In molecular docking studies, receptor grid construction resulted in more reliable ligand posture scoring. As a result, based on the previously acquired binding site residues, we constructed a receptor grid for the selected PDE5 protein in order to obtain a more exact scoring of our ligand poses. A receptor grid with a box dimension of X = 60.9069, Y = 59.9923, and Z = 45.1707 in angstrom (Å) was developed and used for molecular docking research.

### 3.4. Molecular Docking

The structure-based molecular screening method was used to investigate the optimum intermolecular interaction between the target protein (PDE5, PDB ID: 1udt) and bioactive molecules. The efficacy of the 28 selected bioactive molecules and one standard PDE5 inhibitor (sildenafil) was evaluated against the PDE5 protein using the PyRx tool in the AutoDock Vina program. Four bioactive molecules were shown to have a better binding affinity (>−12.30 kcal mol^−1^) against the target protein, PDE5. Based on the molecular docking studies, the binding affinity of the bioactive molecules were viewed to be distributed, ranging from −1.50 to −12.30 kcal mol^−1^, as shown in Table 2. The four top binding bioactive molecules (Bufadienolide (−12.30 kcal mol^−1^), Stigmasterol (−11.40 kcal mol^−1^), Isovitexin (−11.20 kcal mol^−1^), and Apigetrin (−11.20 kcal mol^−1^)) were selected for further work. Further, we used a standard PDE5 inhibitor (sildenafil) (−9.80 kcal mol^−1^) as the control, because it is a widely recommended drug for ED.

### 3.5. Interpretation of Receptor–Ligand Interactions

The BIOVIA discovery studio visualizer tool was used to visualize the formed bonding/interactions between the bioactive molecules and standard drug. The results indicate that the bioactive molecule bufadienolide (BUF) showed the best docking values with the target PDE5, and the binding energy was −12.30 kcal mol^−1^. BUF formed nine hydrophobic interactions: (TYR612A (3.53Å), ASN662A (3.77Å), LEU725A (3.48 Å), LEU725A (3.72Å), VAL782A (3.29Å), PHE786A (3.39Å), PHE820A (3.45Å), PHE820A (3.67Å), and PHE820A (3.70Å)) and two hydrogen bonding (GLN775A (2.23Å) and GLN817A (2.97Å)) with the target protein PDE5, as shown in Figure 3a,b and Appendix A. The bioactive molecule stigmasterol (STI) binds best with the target protein PDE5, and the binding energy was −11.40 kcal mol^−1^. STI made contact with nine hydrophobic interactions (LEU725A (3.09 Å), LEU765A (3.74Å), LEU725A (3.58Å), ALA767A (3.70Å), ILE768A (3.43Å), ILE778A (3.97Å), VAL782A (3.40Å), PHE786A (3.62Å), and PHE820A (3.69Å)) and three hydrogen bonding (ASP724A (2.19Å), ASP724A (2.03Å), and LEU725A (3.23Å)), as depicted in Figure 3c,d and Appendix A. The molecule Isovitexin (ISV) also best docked with the PDE5 protein, having a binding energy of −11.20 kcal mol^−1^. Further, ISV formed four hydrophobic interactions (LEU765A (3.87Å), ALA767A (3.95Å), ILE768A (3.79Å), and VAL782A (3.69Å)) and eleven hydrogen bonding (TYR612A (2.88Å), HIS613A (3.58Å), ASN661A (1.91 Å), ASN662A (3.52Å), ASN662A (3.06Å), THR723A (2.54Å), ASP724A (2.03Å), LEU725A (1.80Å), ASP764A (2.16 Å), ALA767A (3.09 Å), and GLN775A (2.06Å)), as shown in Figure 3e,f and Appendix A. In the case of the bioactive molecule Apigetrin (APT), it showed a better interaction with the target protein PDE5, and the binding energy was −11.20 kcal mol^−1^. APT exhibited three hydrophobic interactions (LEU725A (3.85Å), PHE820A (3.66Å), and PHE820A (3.86Å)); nine hydrogen bonding (TYR612A (1.77Å), TYR612A (2.18Å), HIS613A (2.63Å), ASP654A (3.03Å), HIS657A (3.12Å), ASN662A (2.61Å), GLU682A (2.94Å), HIS685A (2.09Å), and ASP724A (2.61Å)); and two π-stacking ((PHE786A (5.46Å) and PHE820A (4.04Å)) and π-carbon interactions (HIS613A (5.49Å)) with the target protein PDE5, as presented in Figure 3g,h and Appendix A. For the standard PDE5 inhibitor, four hydrophobic bonds were observed with sildenafil (STD) (TYR612A (3.59Å), ILE813A (3.74Å), GLN817A (3.83Å), and PHE820A (3.91Å)); one hydrogen bonding (HIS613A (2.22Å)); and one π-Stacking (PHE786A (5.34Å)) with the target protein PDE, as presented in Figure 3i,j and Appendix A.

### 3.6. Pharmacokinetics and Physicochemical Properties of Bioactive Compounds

Physiologically based pharmacokinetic modeling software tools are increasingly being employed during the drug development process to predict pharmacokinetics and physicochemical characteristics of bioactive molecules or dosage forms [53]. The ability to include changing physiological parameters in mechanically predicting drug disposition is based on this knowledge. Thus, the pharmacokinetics and physicochemical properties of the top binding scored bioactive molecules from *M. pudica* Linn. and standard PDE5 inhibitor (sildenafil) were predicted through the SwissADME online tool, and the observed results are presented in Table 3. From Table 3, the selected top binding scored molecules—BUF (molecular weight: 354.53 g mol^−1^), APT (molecular weight: 432.38 g mol^−1^), and ISV (molecular weight: 432.38 g mol^−1^)—were found to contain one violation of Lipinski’s rule of five, and STI (molecular weight: 464.38 g mol^−1^) and STD (molecular weight: 666.70 g mol^−1^) were observed with two violations of Lipinski’s rule of five. The observed numbers of violations were due to higher molecular weight (>350 g mol^−1^). The polar surface areas of the top binding scored compounds were 30.21 Å^2^ BUF, 210.51 Å^2^ STI, 181.05 Å^2^ ISV, and 170.05 Å^2^ APT. The obtained results revealed that, among the selected four top binding scored bioactive molecules, BUF alone showed a significant gastrointestinal absorption (GI) property, and the other three selected molecules (APT, ISV, and STI), including the STD drug, have low gastrointestinal (GI) absorption properties. Naturally, higher gastrointestinal absorption of the molecule (BUF) leads to enhanced bioavailability. As a result, BUF may be better absorbed in the gastrointestinal tract when taken orally. In addition, the higher the number of H-bonds, the more likely they are engaged in protein ligand binding. Based on the observed results, the bioavailability score for all top binding scored bioactive molecules demonstrated a positive bioavailability score, BUF (+0.55), APT (+0.55), ISV (+0.55), STI (+0.17), and STD (+0.11). Based on this information, all the selected top binding scored bioactive molecules were predicted to have a better chance as a possible drug-like candidate for penile erection. The synthetic accessibility score for all selected top binding scored bioactive molecules with the standard drug sildenafil (PDE5 inhibitor) were found to be >4, which indicated that all the bioactive molecules can be synthesized in a laboratory.

The graphical illustrations of the drug-likeness of the selected top binding scored molecules are presented in Figure 4. The pink area within the hexagon of this graphical representation is an optimal range for bioactive molecules. The recommended range for the drug-like molecule was saturation (SATU): fraction of carbons in the sp^3^ hybridization not less than 0.25, insolubility (INSOLU): log S not higher than 6, hydrophobicity (LIPO): XLOGP3 between −0.7 and +5.0, rotatable bonds (FLEXI): no more than 9 rotatable bonds, molecular weight (SIZE): between 150 and 500 g mol^−1^, and polar surface area (POLAR): TPSA between 20 and 130 Å^2^).

In addition, the pharmacokinetics features of chosen top-scoring bioactive molecules and the standard drug sildenafil were investigated using the boiled egg model, as presented in Figure 5. The boiled egg model provides a quick and reliable way to predict passive gastrointestinal absorption and blood–brain barrier penetration of bioactive molecules, which is useful in the drug design and development process. In Figure 5, the bioactive molecule BUF, found in albumin (white region), revealed better absorption in the gastrointestinal tract. The remaining three bioactive molecules are outside of the boiled egg region using a conventional medication of PDE5 inhibitor. Based on the aforementioned findings, it can be interpreted that the bioactive molecule, BUF, has adequate potential as a drug candidate for prolonged penile erection before and during intercourse.

### 3.7. Analysis of Toxicity

Poor pharmacokinetic and toxic properties are a big issue in drug discovery and development. The pkCSM web-based tool was used to predict the toxicity profile of bioactive molecules. Drug-induced human ether-à-go-go-related gene (hERG) toxicity, AMES toxicity, LD_50_, hepatotoxicity, skin sensitization, *Tetrahymena pyriformis* (TP) toxicity, and minnow toxicity were all discovered by the server and are reported in Table 4.

### 3.8. Molecular Dynamics (MD) Simulation

Although protein–ligand docking has been widely used and has proved to be effective, it only provides a static picture of the binding posture of the ligand in the active region of the receptor. With the integration of Newton’s equations of motion, molecular dynamics must be used to model the atomic dynamics in the system throughout time. In order to better understand the unbound APO protein form of the PDE5 protein, MD simulation lasting 100 ns was performed on the top binding scored four test complexes (ligand-receptor complex) and one standard complex derived from docking studies, namely: BUF, APT, ISV, STI, and STD. MD trajectories analysis was used to determine the stability and fluctuations of these complexes using RMSD, RMSF (Root Mean Square Fluctuation), RGY (Radius of gyration), and SASA (Solvent Accessible Surface Area) of the receptor atoms.

RMSD is an essential measure for analyzing the MD trajectory equilibration and checking the stability of complicated systems throughout the simulation process. To measure the differences in the structural confirmation, the RMSD of the protein backbone atoms was plotted versus time. During the initial period of the simulation, a minor fluctuation was observed for all the complexes and APO protein, later becoming stable throughout the simulation period of 100 ns (Figure 6a). The average backbone RMSD of all complexes were computed. Complexes BUF, ISV, STI, and STD showed high stability, and the APT complex showed lower stability than the APO protein (Table 5).

RMSF is another critical metric to consider while simulating the stability and flexibility of complex systems. The RMSF was used to determine how amino acid residues of a target protein changed their behavior upon binding to a ligand. The RMSF values for the protein’s Cα atoms were computed and shown against the residues. In all complexes, the amino acid residues exhibited very little variation during the simulation (Figure 6b). The average backbone RMSF were calculated, which indicated that all the complexes showed less fluctuation when compared with the APO protein (Table 5). These findings indicate that the binding of ligands had no discernible influence on the flexibility of the protein’s residues.

Additionally, the complex system radius of gyration (RGY) was analyzed. RGY is the root mean square distance between the protein’s atoms and the rotation axis. It is one of the critical parameters that indicates the overall change in the compactness and dimensions of the protein structure throughout the simulation. Increased RGY values indicate that the protein is less compact and flexible, while low values indicate that the protein is very compact and inflexible. RGY values of backbone atoms of protein were plotted against time to examine the changes in structural compactness. Protein and protein–ligand complexes showed a gradual decrease in the RGY value throughout the simulation, which revealed that the test molecules induced no major structural changes in the protein (Figure 6c).

Furthermore, the SASA analysis for all complexes was carried out. SASA is a significant parameter for examining the degree of receptor exposure to the surrounding solvent molecules during simulation. In general, ligand binding may cause structural changes in the receptor, causing the region in contact with the solvent to alter. SASA values of protein were plotted against time to estimate the changes in surface area. The protein and protein–ligand complexes showed a gradual decrease in the solvent accessible surface area throughout the simulation, which revealed that the surface area of protein in both complexes shrunk during the simulation (Figure 6d).

APT generated a high number of H-bonds with the receptor protein, with a maximum of five bonds formed across numerous time frames. Throughout the simulation duration, approximately two hydrogen bonds were formed consistently, indicating the complex’s stability (Figure 6e). The consistency was maintained in the formation of two hydrogen bonds for the BUF complex. The consistency in establishing three hydrogen bonds with a maximum of seven bonds at various time periods was maintained for the ISV complex. The consistency was maintained in the formation of two hydrogen bonds for the STI complex. The consistency in establishing four hydrogen bonds with a maximum of five bonds at a particular time period was maintained for the STD complex. This indicates that the top phytochemicals chose a higher affinity for the target protein.

### 3.9. MM/PBSA—Binding Free Energy Analysis

The binding free energy (ΔG bind) for all complexes was determined using the MM/PBSA technique for the final 20 ns (80–100 ns) of the simulated trajectories with dt 1000 frames. Complexes BUF, APT, ISV, and STI showed more binding free energy in comparison with the STD (Table 6). These low negative free binding energies indicate that the test ligands have a strong affinity for binding to the target protein. The per-residue energy decomposition of each complex was assessed (Appendix A). The main contributions with binding energy higher than −5.0 KJ.mol^−1^ observed for the APT complex were PHE-820, PHE-786, LEU-804, VAL-782, TYR-612, and LEU-765; for the BUF complex were PHE-820, GLN-817, MET-816, TYR-664, and LEU-765; for the ISV complex were PHE-820, VAL-782, PHE-786, TYR-612, LEU-765, MET-816, and GLN-817; for the STI complex were PHE-820, LEU-765, and VAL-782; and for the STD complex were PHE-820, PHE-786, TYR-664, and TYR-676. The interaction analysis and their individual energy contributions indicated that most of the interactions were consistent.

### 3.10. Density Functional Theory

The frontier-orbital energies (highest occupied molecular orbital (HOMO) & lowest unoccupied molecular orbital (LUMO)) of the bioactive molecules play a vital role in biological functions since HOMO is a basic measure of a compound’s electron-donating capacity. The energies and HOMO–LUMO energy gap of chosen bioactive molecules APT, BUF, ISV, STI, and STD were assessed using the B3LYP level with the 6-311G (d, p) basis set, and the HOMO–LUMO diagram is shown in Table 7. Of the bioactive molecules, STI has the highest HOMO-LUMO energy gap of 5.055062, BUF had an energy gap of 3.702111, APT had an energy gap of 3.388364, and ISV had an energy gap of 3.381833, while the standard drug (sildenafil) had a HOMO-LUMO energy gap of 2.841414. The energy gap measurements revealed that all four selected bioactive molecules are both extremely stable in nature.

## 4. Discussion

The main aim of this work was to find an effective bioactive molecule from *M. pudica* Linn. for the treatment of ED using the in silico molecular modeling approach. The crude extract of *M. pudica* Linn. provides a significant and constant increase in aphrodisiac performance of male mice [54]. However, the molecular mechanisms underlying their actions remain unknown. In this study, molecular modeling tools were used for the screening of molecules from *M. pudica* Linn. based on their binding affinity calculations against the target protein (PDE5). Pharmacokinetics and physicochemical properties of the selected compounds were also evaluated. In silico molecular modeling tools helped narrow down the number of molecules and speed up the drug development processes [55]. In general, computer-aided drug design and discovery was used to anticipate the promising effects of certain natural bioactive molecules, which were later confirmed by in vitro and in vivo studies. Understanding how bioactive molecules bind to, interact with, and inhibit/stimulate a protein (receptor) could help researchers to discover therapeutic solutions for disorders.

In the present study, a graph theoretical network concept was utilized to reconstruct the signaling pathway of the PDE5 protein to explore the roles and responsibilities of various genes, enzymes and proteins involved. The obtained results of centrality calculations (closeness, betweenness, eigenvector, eccentricity, stress, radiality and degree), human phosphodiesterase type 5 (PDE5) enzyme was identified as a target (receptor) for binding bioactive compounds (ligands). PDE5 is an enzyme generally found in the smooth muscle of the corpus cavernosum. Pande reported that the ethanolic extract of roots of *M. pudica* Linn. showed a significant and sustained increase in the aphrodisiac activity of normal male mice, without any adverse effects [30]. On the basis of this study, we hypothesised that bioactive molecules from *M. pudica* Linn. inhibit the PDE5 enzyme as a result of nitric oxide release, which leads to increasing the levels of cGMP (L-arginine–nitric oxide–guanylyl cyclase–cyclic guanosine monophosphate) in the corpus cavernosum. The increasing cGMP in the smooth muscle cells, which is responsible for smooth muscle relaxation and the flow of blood in the corpus cavernosum, prolongs a penile erection. Further, numerous studies have now been explored deciphering the role of oxidative stress and ROS in pathophysiology of ED [10,11,12,56]. Importantly, the reactions of free radicals not only impair cavernosal relaxation but also lead to prolonged penile vasculopathy [57]. In addition, ROS have also been connected with neurogenic, vasculogenic ED, diabetes mellitus, and cardiovascular diseases [57,58]. The currently available drugs for the treatment of ED produce life-threatening toxic effects because PDE5 is expressed throughout the body [59]. In general, bioactive molecules obtained from plants are cellular protective, cost-effective, safe, and are able to reduce/neutralise undesirable harmful effects and maintain the level of ROS (free radical scavenging) in the human body [60]. In this view point, 28 bioactive molecules were selected from *M. pudica* Linn. through the IMPPAT database and previously published articles on the basis of therapeutic importance. All the selected molecules were docked against the PDE5 enzyme and observed binding affinities ranging between −1.5 kcal mol^−1^ to −12.30 kcal mol^−1^. Based on their binding affinities and strong interactions between the target protein with amino acid residues, four bioactive molecules, BUF (−12.30 kcal mol^−1^), STI (−11.40 kcal mol^−1^), ISV (−11.20 kcal mol^−1^), APT (−11.20 kcal mol^−1^) and standard PDE5 inhibitor (sildenafil) (−9.80 kcal mol^−1^) were selected for further experiments.

Pharmacokinetics (ADMET), pharmacodynamics, and physicochemical properties are all connected to phytoconstituents’ effectiveness, which are included in the novel drug discovery process. The study of these mechanisms, which entails the movement of molecules to various physiologic compartments and how they function/affect the use and utility of molecules. The bioavailability of phytoconstituents to target cells, as well as their absorption and metabolism in the human body, are all important factors in increasing bioactivity and body health. However, phytoconstituents with poor pharmacokinetics and toxic properties must be eliminated during the drug discovery process, as it is critical that natural molecules be therapeutically active and have relevant pharmacokinetics properties. At this point, there are numerous phytocompounds to explore, but the physical sample availability is limited. For this reason, computer-aided simulations are a viable alternative to animal experiments. One of the most commonly used drug-likeness properties of phytoconstituents is Lipinski’s rule of five, which predicts a phytocompound’s ability to be orally active in the human system [61]. Under pharmacokinetic properties that influence the permeability of bioactive molecules across biological barriers (i.e., blood brain barrier, membrane permeability, and gastrointestinal absorption), molecular weight and topological polar surface area (TPSA) have been identified [62]. The TPSA is known as the surface area filled by nitrogen and oxygen atoms, as well as the polar hydrogens bonded to these heteroatoms [63]. The permeability of a phytoconstituents through biological barriers decreases when the hydrophilic part of its surface increases as well as higher molecular weight (<500 g mol^−1^) [64]. Solubility, stability owing to extreme pH values in the stomach and metabolism by gut microorganisms also influence the absorption of phytocompounds in the body [65]. Of the selected top binding bioactive molecules, BUF alone presents higher gastrointestinal absorption, while the other three molecules (STI, APT, and ISV) with the standard PDE5 inhibitor (sildenafil) showed less gastrointestinal absorption, which could be attributed to the higher molecular weight of these three molecules. Some phytochemicals are known to be taken into the circulatory system through the small intestine, whereas others are believed to be absorbed by the colon and altered by the gut microbiota before being released back into the bloodstream along with some microbial metabolites [66]. These are known to have potent pharmacological effects. A number of hydrogen bond acceptors/donors of phytoconstituents are known to influence their permeability, in addition to increasing their molecular weight and solubility [67]. Phytocompounds with five or fewer hydrogen bond donors have a high degree of permeability through biological barriers [68]. The bioactive molecules examined in this study, BUF, STI, APT, ISV, and the standard PDE5 inhibitor (sildenafil) have 0, 8, 6, 7, and 5 hydrogen bond donors, respectively, and 2, 12, 10, 10, and 12 hydrogen bond acceptors, respectively, which indicate moderate permeability through biological barriers. Additionally, better oral bioavailability indicates phytocompounds with 10 or fewer rotatable bonds [69]. The human intestinal absorption and blood-brain barrier permeability of selected phytocompounds were determined using the boiled-egg method. In our study, BUF alone found the white region (albumin region) of the boiled-egg, indicating that the molecule has good gastrointestinal absorption. Phytochemicals are naturally derived molecules found in a wide range of plants commonly taken by people and are generally considered harmless. The Food and Drug Administration (FDA) in the United States does not restrict most phytochemicals since they have no known potential toxicity [70].

Toxicology of phytocompounds must be determined in order to investigate their unfavorable harmful effects on humans. It is also an essential stage in the drug discovery process. Despite the fact that animal models are poor indicators of drug safety in humans, they are still being utilized in pharmaceutical and industrial research to predict human toxicity. Since animal research leads to high cost and delay in drug approval, the availability of potentially valuable medications for human use gets delayed. As a result, computer-aided approaches for predicting the toxicity of phytocompounds are recognized as useful [71]. In our study, toxicity tests revealed that the four selected bioactive molecules had no harmful side effects (hERG toxicity, carcinogenicity, skin irritation and exhibited favorable logBB values). The LD_50_ (median fatal dosage) represents the immediate or acute toxicity of the bioactive molecules that were shown to be most effective in this study. As a result, MD simulation was performed on the complexes of these molecules, and the results were compared to those of the APO form of the PDE5 protein. The complexes (protein–ligand) were validated by interpreting the RMSD, RMSF, RGY, SASA and the lead phytochemical complexes were found to be stable during the simulation, which indicates the test complex does not influence the stability and flexibility of protein. Moreover, pre-residue energy contribution data supports the molecular docking interactions. The key amino acids PHE-820 and PHE-786 play a vital role in the native protein function, to which the test drugs and standard drugs bind during molecular docking and dynamics simulation.

## 5. Conclusions

Men’s personal identity and self-efficacy can be highly dependent on sex, and when they are unable to have an erection or have sex in the same way they used to, it can place stress on their relationships. Many traditional medicines, in particular plants and their molecules, have been shown to be highly effective without causing any harmful effects, improve health and are cost effective, while also enhancing the quality of life. This study explored the effectiveness of bioactive molecules from *M. pudica* Linn. in inhibiting the effect of the PDE5 enzyme through in silico molecular docking and molecular dynamics simulation studies. PDE5 is an important enzyme found in the smooth muscle of the corpus cavernosum, and it can be selectively targeted and its activity inhibited. Through a detailed molecular docking analysis of bioactive molecules, four leading bioactive molecules (Bufadienolide, Stigmasterol, Isovitexin, and Apigetrin) were selected based on their least/better binding affinities to the PDE5 enzyme than the standard PDE5 inhibitor (sildenafil). In addition to this, studies of molecular dynamics simulation, pharmacokinetics and physicochemical properties confirm the safety profiles and stability of PDE5–ligands complexes of the four chosen bioactive molecules. This further confirms the effectiveness of the bioactive molecules. Additional in vitro and in vivo animal studies are warranted to evaluate the aphrodisiac performance of these bioactive molecules.

## Figures and Tables

**Figure 1 molecules-27-03799-f001:**
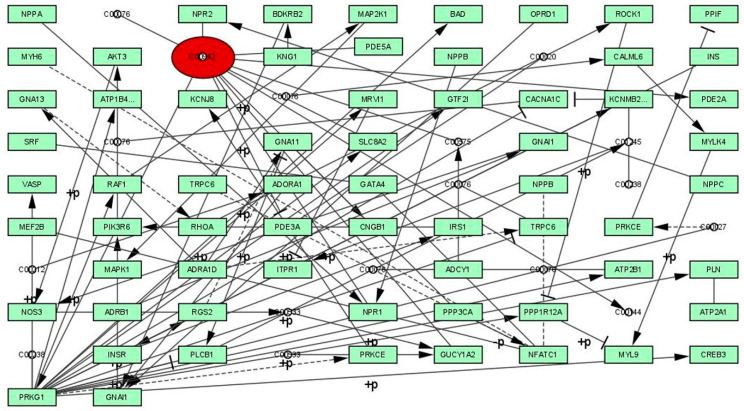
The signaling pathway of phosphodiesterase type 5 (PDE5).

**Figure 2 molecules-27-03799-f002:**
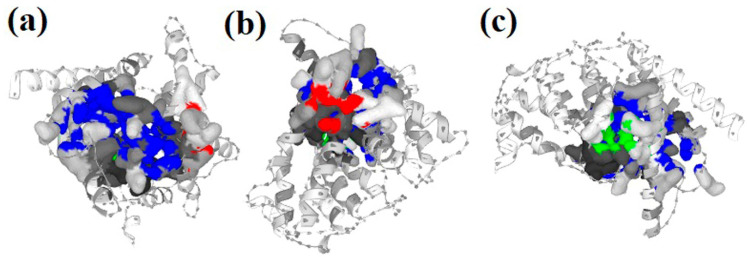
The PrankWeb tool is used to predict the binding pockets and correspondence binding sites of the PDE5 enzyme. Three binding pockets were predicted with different colors. The first pocket is colored in blue with a pocket score of 67.90, 36 amino acids, and a probability score of 0.990 (**a**). The second pocket is in red, with a pocket score of 1.23, 7 amino acids, and a probability score of 0.012 (**b**). The third pocket is in green with a pocket score of 1.15, 9 amino acids, and a probability score of 0.010 (**c**).

**Figure 3 molecules-27-03799-f003:**
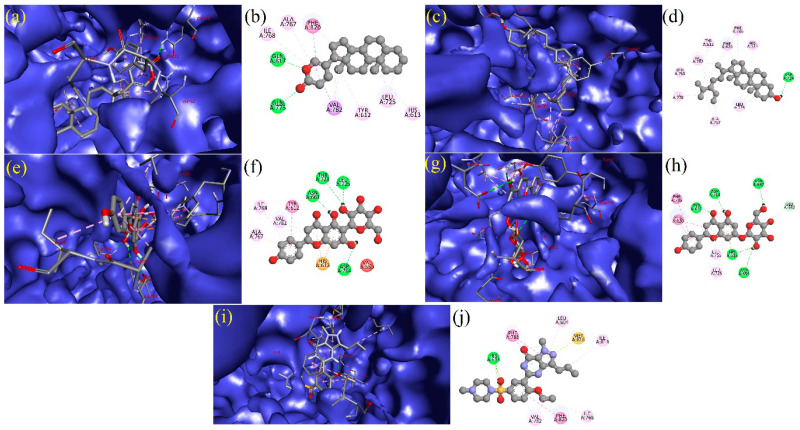
Depicted the interaction between the compound Bufadienolide and PDE5. Left side representing the 3D (**a**) and the right side representing the 2D complex of the PDE5–Bufadienolide interaction (**b**). Depicted the interaction between the compound Stigmasterol and PDE5. Left side representing the 3D (**c**) and the right side representing the 2D complex of the PDE5–Stigmasterol interaction (**d**). Depicted the interaction between the compound Isovitexin and PDE5. Left side representing the 3D (**e**) and the right side representing the 2D complex of the PDE5–Isovitexin interaction (**f**). Depiction of the interaction between the compound Apigetrin and PDE5. Left side representing the 3D (**g**) and the right side representing the 2D complex of the PDE5–Apigetrin interaction (**h**). Depiction of the interaction between the compound sildenafil and PDE5. Left side representing the 3D (**i**) and the right side representing the 2D complex of the PDE5–sildenafil interaction (**j**).

**Figure 4 molecules-27-03799-f004:**
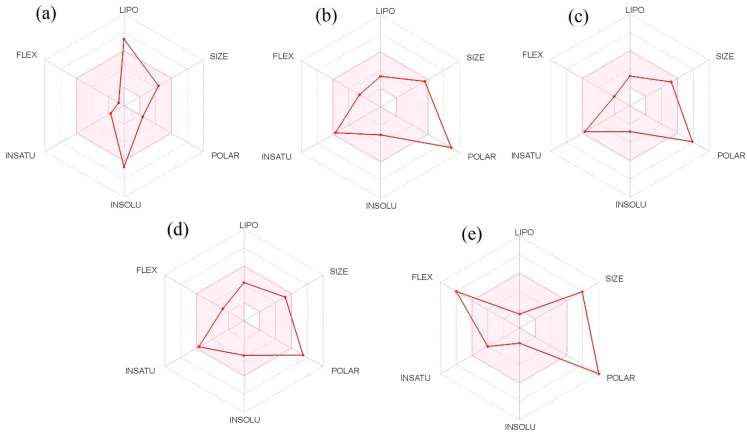
Bioavailability radar plot for oral bioavailability of top binding scored bioactive molecules: Bufadienolide (**a**), Stigmasterol (**b**), Isovitexin (**c**), Apigetrin (**d**), and standard PDE5 inhibitor sildenafil (**e**). The pink area exhibits the optimal range for each of the properties (Lipophilicity as XLOGP3 between −0.7 and +5.0, size as molecular weight between 150 and 500 g mol^−1^, polarity as TPSA (topological polar surface area) between 20 and 130 Å^2^, insolubility in water by log S scale not higher than 6, saturation as per fraction of carbons in the sp3 hybridization not less than 0.25, and flexibility as per rotatable bonds no more than 9).

**Figure 5 molecules-27-03799-f005:**
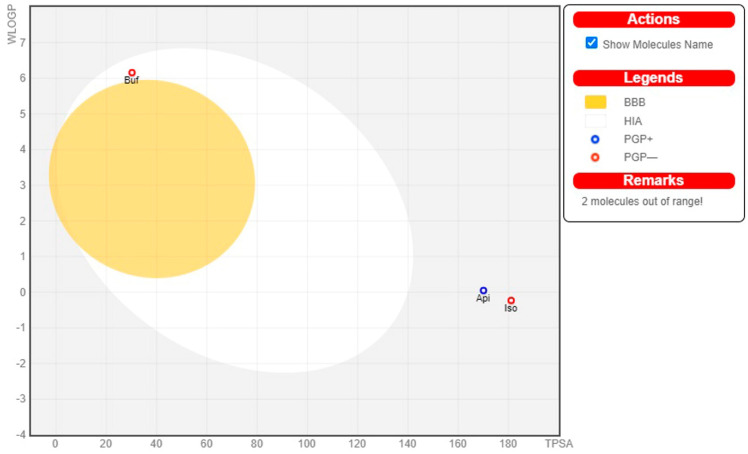
The egg boiled model for the top scored bioactive molecules and standard PDE5 inhibitor sildenafil. The egg boiled represents the intuitive evaluation of passive gastrointestinal absorption (HIA) as the white part and blood–brain penetration (BBB) as the yellow part, while the substrates (PGP+) and non-substrates (PGP−) of the permeability glycoprotein (PGP) are represented by blue and red color circles, respectively, of the selected top binding scored bioactive molecules and standard PDE5 inhibitor sildenafil in the WLOGP versus TPSA graph. The grey region is the physicochemical space of compounds predicted to exhibit high intestinal absorption.

**Figure 6 molecules-27-03799-f006:**
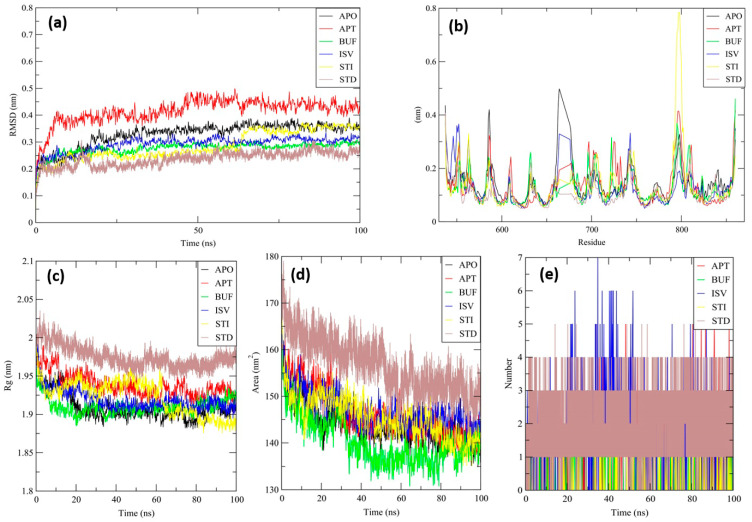
RMSD study plot for 100-ns MD simulation of PDE5-APO (Black), PDE5-BUF (Green), PDE5-STI (Yellow), PDE5-ISV (Blue), PDE5-APT (Red), and PDE5-STD drug sildenafil (Brown) (**a**). RMSF study plot for 100-ns MD simulation of PDE5-APO (Black), PDE5-BUF (Green), PDE5-STI (Yellow), PDE5-ISV (Blue), PDE5-APT (Red), and PDE5-STD drug sildenafil (Brown) (**b**). Radius of gyration study plot for 100-ns MD simulation of PDE5-APO (Black), PDE5-BUF (Green), PDE5-STI (Yellow), PDE5-ISV (Blue), PDE5-APT (Red), and PDE5-STD drug sildenafil (Brown) (**c**). Solvent accessible surface area study plot for 100-ns MD simulation of PDE5-APO (Black), PDE5-BUF (Green), PDE5-STI (Yellow), PDE5-ISV (Blue), PDE5-APT (Red), and PDE5-STD drug sildenafil (Brown) (**d**). Intermolecular hydrogen bonding study plot for 100-ns MD simulation of PDE5-APO (Black), PDE5-BUF (Green), PDE5-STI (Yellow), PDE5-ISV (Blue), PDE5-APT (Red), and PDE5-STD drug sildenafil (Brown) (**e**).

**Table 1 molecules-27-03799-t001:** The results of the threshold parameter values of the phosphodiesterase type 5 enzyme network analysis.

Gene	Betweenness	Closeness	Degree	Eccentricity	Eigen Vector	Radiality	Stress
PRKG1	4355.8	0.004524887	20	0.111111111	0.664578142	10.96341	6142
C00942	3193.75	0.004166667	10	0.125	0.241942799	10.73171	4126
NFATC1	805.6666667	0.002304147	6	0.076923077	0.005206962	8.365854	1684
GNA11	1051.166667	0.003194888	4	0.090909091	0.084715899	9.841463	2202
GNAI1	718.5	0.001883239	4	0.076923077	0.0000377	7.378049	856
PPP1R12A	752.0666667	0.003571429	3	0.1	0.1554146	10.2439	964
NPR1	247.75	0.003267974	3	0.111111111	0.056256814	9.926829	350
GUCY1A2	1814	0.003663004	3	0.125	0.056401272	10.32927	2366
KCNMB2	110	0.003448276	3	0.1	0.189210899	10.12195	176
ADRA1D	185.8333333	0.003144654	3	0.090909091	0.054578549	9.780488	228
PLCB1	673	0.003717472	3	0.1	0.16797582	10.37805	1314
NOS3	1596.5	0.002777778	3	0.1	0.002934907	9.268293	2020
PIK3R6	1192.5	0.002150538	3	0.083333333	0.000154	7.987805	1476
TRPC6	311.1	0.003597122	3	0.1	0.159374597	10.46341	400
CACNA1C	86.2	0.003436426	3	0.1	0.189210899	10.10976	176
MYL9	373.4	0.002915452	2	0.090909091	0.034521634	9.47561	516
ITPR1	2	0.002873563	2	0.090909091	0.040978905	9.414634	2
NPR2	148	0.003367003	2	0.111111111	0.053617712	10.03659	176
MRVI1	125	0.003472222	2	0.1	0.149354275	10.14634	136
RGS2	510	0.003703704	2	0.1	0.158612643	10.36585	1134
RHOA	19.03333333	0.002557545	2	0.083333333	0.010336014	8.890244	32
ROCK1	130.5	0.002915452	2	0.090909091	0.03508655	9.47561	142
C01245	21	0.00295858	2	0.090909091	0.044232116	9.536585	24
PPP3CA	909.8333333	0.002688172	2	0.083333333	0.019035121	9.121951	1918
GATA4	148	0.002012072	2	0.071428571	0.001153931	7.597561	286
PLN	148	0.003484321	2	0.1	0.147279271	10.15854	176
RAF1	292	0.003558719	2	0.1	0.147604105	10.23171	348
MAP2K1	148	0.002832861	2	0.090909091	0.032711014	9.353659	176
SLC8A2	86.2	0.003460208	2	0.1	0.147279271	10.13415	176
ATP2B1	86.2	0.003436426	2	0.1	0.147279271	10.10976	176
AKT3	1271	0.002415459	2	0.090909091	0.000654	8.609756	1580
C00533	1663	0.003095975	2	0.111111111	0.012560447	9.914634	2100
C00575	148	0.003389831	2	0.111111111	0.053617712	10.06098	176
PDE3A	148	0.003355705	2	0.111111111	0.053617712	10.02439	176
PRKCE	148.5	0.002793296	2	0.090909091	0.032711014	9.487805	176
PRKCE	148.5	0.003367003	2	0.1	0.147279271	10.23171	176
MYLK4	231.8	0.002439024	2	0.083333333	0.007667322	8.658537	348
CALML6	86.2	0.002079002	2	0.076923077	0.00169918	7.792683	176
IRS1	292	0.001872659	2	0.076923077	0.0000341	7.146341	348
INSR	148	0.001666667	2	0.071428571	0.00000756	6.341463	176
GNAI1	148.5	0.002309469	2	0.090909091	0.00065	8.573171	176
BDKRB2	148	0.001683502	2	0.071428571	0.00000837	6.414634	176
ADORA1	148	0.001666667	2	0.071428571	0.00000837	6.341463	176
GNA13	32.53333333	0.002617801	2	0.083333333	0.013741295	9	50
C00027	255	0.003472222	2	0.1	0.147604105	10.14634	348

**Table 2 molecules-27-03799-t002:** Bioactive molecules from *Mimosa pudica* L. and their binding affinity against the phosphodiesterase type 5 (PDE5) enzyme.

S. No	Compound ID	Bioactive Molecule	Binding Affinity
1.	94477	Mimosinamine	−5.50
2.	125409	Beta-D-xylopyranose	−5.70
3.	190359	Mimosinic Acid	−6.00
4.	370	Gallic Acid	−6.20
5.	951	Dl- Norepinephrine	−6.30
6.	164619	D-Pinitol	−6.40
7.	3862	Mimosine	−6.60
8.	1153	DL-tyrosine	−6.60
9.	71684438	Octadecadienoicacid	−6.80
10.	94715	D-Glucopyranuronate	−6.80
11.	5280441	Vitexin	−7.00
12.	440473	L-Mimosine	−7.10
13.	5281166	Jasmonic Acid	−7.10
14.	5375199	Abscisic acid	−7.30
15.	100927206	Mimopudine	−7.80
16.	5280489	Beta-Carotene	−8.60
17.	5281679	Methylquercetin	−9.30
18.	64971	Betulinic Acid	−9.70
19.	222284	Beta-sitosterol	−10.30
20.	5490064	Avicularin	−10.50
21.	5281675	Orientin	−10.60
22.	114776	Isoorientin	−10.70
23.	70698280	Cassiaoccidentalin B	−10.80
24.	5280804	Isoquercitrin	−10.90
25.	162350	Isovitexin	−11.20
26.	5280704	Apigetrin	−11.20
27.	5280794	Stigmasterol	−11.40
28.	46173848	Bufadienolide	−12.30
Standard drug			
29.	135398744	Sildenafil 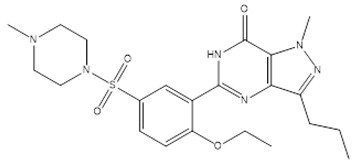	−9.8

**Table 3 molecules-27-03799-t003:** Pharmacokinetics and physicochemical parameters of selected top binding scored bioactive molecules and the standard PDE 5 inhibitor.

Parameters	Bufadienolide (CID:46173848)	Stigmasterol (CID:5280794)	Apigetrin (CID:5280704)	Isovitexin (CID:162350)	Sildenafil (CID: 135398744)
Formula	C_24_H_34_O_2_	C_21_H_20_O_12_	C_21_H_20_O_10_	C_21_H_20_O_10_	C_22_H_30_N_6_O_4_S
MW (g mol^−1^)	354.53	464.38	432.38	432.38	474.58
Num. heavy atoms	26	33	31	31	33
Num. arom. heavy atoms	6	16	16	16	15
Fraction Csp3	0.79	0.29	0.29	0.29	0.50
Num. rotatable bonds	1	4	4	3	7
Num. H-bond acceptors	2	12	10	10	8
Num. H-bond donors	0	8	6	7	1
Molar Refractivity	107.49	110.6	106.11	106.61	134.56
TPSA (Å^2^)	30.21	210.51	170.5	181.5	121.80
Solubility class	Poorly Soluble	Soluble	Soluble	Soluble	Soluble
GI absorption	High	Low	Low	Low	High
BBB permeation	NO	NO	NO	NO	NO
Violation of Lipinski’s rule of five	1	2	1	1	1
Violation of Veber rule	0	0	0	0	0
Bioavailability Score	0.55	0.17	0.55	0.55	0.55
Synthetic accessibility	5.60	5.32	5.12	4.99	3.95

**Table 4 molecules-27-03799-t004:** List of the drug-induced hERG inhibition, AMES toxicity, carcinogens, *Tetrahymena pyriformis* (TP) toxicity, rat acute toxicity (LD_50_ in mol kg^−1^), and skin sensitization along with Minnow toxicity of selected top binding scored molecules.

Compound	AMES Toxicity	Max. Tolerated Dose (Human)	hERG Inhibition	LD50	Hepatotoxicity	Carcinogenicity	Cytotoxicity	Skin Sensitization	*T. pyriformis* Toxicity	Minnow Toxicity
Bufadienlide	NO	−0.087	No	2.638	No	No	No	NO	0.417	−1.896
Stigmasterol	NO	0.508	NO	2.624	NO	No	No	NO	0.285	2.706
Apigetrin	NO	0.698	NO	2.442	NO	No	No	NO	0.285	1.131
Isovitexin	YES	1.036	NO	3.034	NO	No	No	NO	0.285	2.803
Sildenafil	NO	0.209	NO	2.655	YES	No	No	NO	0.286	0.889

**Table 5 molecules-27-03799-t005:** Molecular dynamics simulation for the selected top binding scored bioactive molecules.

S. No	Protein–ligand Complex	Average Backbone RMSD (nm)	Average Backbone RMSF (nm)
1	APO	0.328935 ± 0.044482	0.140706 ± 0.068559
2	API	0.415632 ± 0.044550	0.135675 ± 0.072969
3	BUF	0.276698 ± 0.022447	0.126093 ± 0.060376
4	ISV	0.292700 ± 0.026475	0.121282 ± 0.060832
5	STI	0.285493 ± 0.052889	0.134828 ± 0.102423
6	STD	0.236710 ± 0.027702	0.111075 ± 0.052376

**Table 6 molecules-27-03799-t006:** MM/PBSA energy for top ranked complex.

S. No	Protein–ligand Complex	ΔGbind (kJ mol^−1^)
1	API	−212.407 ± 17.541
2	BUF	−233.376 ± 14.471
3	ISV	−211.953 ± 10.191
4	STI	−210.678 ± 12.676
5	STD	−164.117 ± 16.451

**Table 7 molecules-27-03799-t007:** EHOMO and ELUMO and ΔE values of selected top binding scored bioactive molecules and standard PDE 5 inhibitor.

Compound	HOMO		LUMO		Energy Gap
BUF	−9.27337	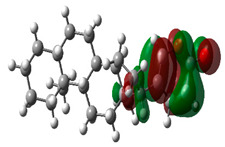	−5.57126	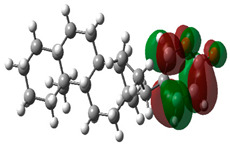	3.702111
STI	−9.84236	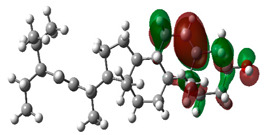	−4.7873	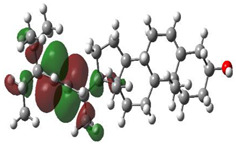	5.055062
API	−8.94139	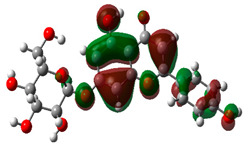	−5.55303	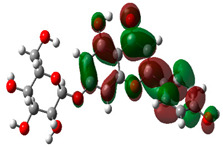	3.388364
ISO	−8.93432	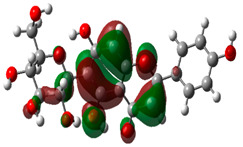	−5.55249	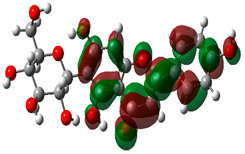	3.381833
STD	−8.35254	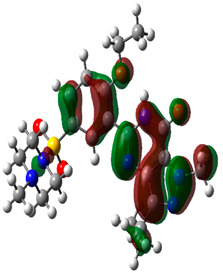	−5.51112	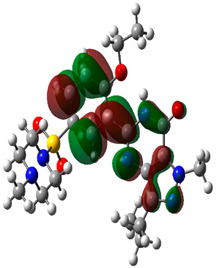	2.841414

## Data Availability

The original contributions presented in the study are included in the article, and further inquiries can be directed to the corresponding author/s.

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
