# Peer review of "Aphrodisiac Performance of Bioactive Compounds from Mimosa pudica Linn.: In Silico Molecular Docking and Dynamics Simulation Approach"

_molecules, 2022, doi:10.3390/molecules27123799_

Round 1

Reviewer 1 Report

Dear Sir

The authors presented Aphrodisiac performance of bioactive compounds from Mimosa pudica L.: In silico molecular docking and dynamics simulation approach. The work is interested and can be accepted but the following comments must be considered before production

My comments

  • Aim of the work should be stated clearly in introduction
  • Introduction should be more concise
  • Resolution of figure 2 should be enhanced
  • For data set of DFT calculation you have to add references

 Frisch, M.J.; Trucks, G.W.; Schlegel, H.B.; Scuseria, G.E.; Robb, M.A.; Cheeseman, J.R.; Scalmani, G.; Barone, V.; Petersson, G.A.; Nakatsuji, H.; et al. Gaussian 09, Revision, D.01; Gaussian, Inc.: Wallingford, CT, USA, 2013; Available online: https://gaussian.com/g09citation (accessed on 1 March 2022).; International Journal of Molecular Sciences 2022, 23, no. 7: 3994.

  • The current study should be compared with other related studies in literature
  • The authors must revise language of the manuscript before publication and the whole article must be adjusted based on journal style.

Author Response

Reviewer 1

The authors presented Aphrodisiac performance of bioactive compounds from Mimosa pudica L.: In silico molecular docking and dynamics simulation approach. The work is interested and can be accepted but the following comments must be considered before production.

My comments

  1. Aim of the work should be stated clearly in introduction

Answer: Thank you for your valuable suggestion, now aim of the experiment included in the introduction in our revised manuscript.

  1. Introduction should be more concise

Answer: Now introduction part was reduced the revised manuscript.

  1. Resolution of figure 2 should be enhanced

Answer: Thank you for your view points, the resolution of Figure 2 was improved in the revised manuscript

  1. For data set of DFT calculation you have to add references

Frisch, M.J.; Trucks, G.W.; Schlegel, H.B.; Scuseria, G.E.; Robb, M.A.; Cheeseman, J.R.; Scalmani, G.; Barone, V.; Petersson, G.A.; Nakatsuji, H.; et al. Gaussian 09, Revision, D.01; Gaussian, Inc.: Wallingford, CT, USA, 2013; Available online: https://gaussian.com/g09citation (accessed on 1 March 2022).; International Journal of Molecular Sciences 2022, 23, no. 7: 3994.

Answer: The mentioned references were incorporated in our revised manuscript. 

  1. The current study should be compared with other related studies in literature

Answer: Thank you for your valuable suggestion, the comparison between other articles were incorporated in our revised manuscript.

  1. The authors must revise language of the manuscript before publication and the whole article must be adjusted based on journal style.

Answer: Sorry for the inadvertent language errors in the manuscript. Now, we have thoroughly checked and corrected all the language errors. 

Reviewer 2 Report

Table 1 seems to include Mimosa pudica L. compounds together with drugs that were used as a reference. Could it be divided into two parts – one with extract compounds and second with reference drugs?

“From the IMPPAT database and published articles, three-dimensional structures of 227 28 bioactive molecules from the target plant M. pudica L “ - Could Authors attach results of their search in IMPPAT for M. pudica? There is no Sildenafil or similar compound among extract compounds there. Only 13 compounds are there listed for M. pudica. Also, there is no such disease like male erectile dysfunction associated with M. pudica listed in IMPPAT. Could Authors provide the source of the information other than IMPPAT which is indeed the basis of their study? Ref. 27 provides only associations of M. pudica with neurodegenerative diseases and some data is on its antiparasitic activity. Precise reference should be provided for each of these four extract compounds mentioned in Conclusions: Bufadienolide, Stigmasterol, Isovitexin, and Apigetrin and linking the M. pudica extract with PDE5.

Minor misspellings such as: „Finding affinity” in Table 1 instead of binding.

Author Response

Reviewer 2

Table 1 seems to include Mimosa pudica L. compounds together with drugs that were used as a reference. Could it be divided into two parts – one with extract compounds and second with reference drugs?

Answer: We appreciate the reviewer for their critical review of our manuscript. In the revised manuscript standard drug was separated in Table 1.

“From the IMPPAT database and published articles, three-dimensional structures of 227 28 bioactive molecules from the target plant M. pudica L “ - Could Authors attach results of their search in IMPPAT for M. pudica? There is no Sildenafil or similar compound among extract compounds there. Only 13 compounds are there listed for M. pudica. Also, there is no such disease like male erectile dysfunction associated with M. pudica listed in IMPPAT. Could Authors provide the source of the information other than IMPPAT which is indeed the basis of their study? Ref. 27 provides only associations of M. pudica with neurodegenerative diseases and some data is on its antiparasitic activity. Precise reference should be provided for each of these four extract compounds mentioned in Conclusions: Bufadienolide, Stigmasterol, Isovitexin, and Apigetrin and linking the M. pudica extract with PDE5.

Answer: We appreciate the reviewer for their critical review of our articles. In the revised manuscript included another one column for literature source of the bioactive compounds. Yes sir, Mimosa pudica Linn. in silico work is not yet published, this is the first work, and aphrodisiac performance of Mimosa pudica reported in vivo animal model. Further, Ref. 27 was removed and appropriate references were included in our revised manuscript.

Minor misspellings such as: „Finding affinity” in Table 1 instead of binding.

Answer: Spelling mistakes are corrected in our revised manuscript. 

Reviewer 3 Report

Comments on molecules-1735970:

The current paper reports a combined experimental and computational screening of bioactive compounds from Mimosa pudica L. The protein-binding performance of four top-binding compounds and their toxicity behavior were checked further to validate their potential use. Although many findings are biologically relevant and could be helpful to the molecules audience, significant improvements are required in many parts of the paper, especially the simulation part. Although I would like to consider a revised version for publication, they need to re-do the whole simulation part carefully. Please note that the reviewers’ comments are supportive and are provided to improve the quality of the research rather than criticizing anything purposedly, and the authors should take these suggestions carefully in the intended spirit.

Many abbreviations defined in the manuscript were not mentioned later. For example, ED defined in the first sentence and SPC defined in Section 2.7 are never mentioned later. In that case, these abbreviations should not be defined at all. The authors should check this issue in the whole manuscript carefully.

Inconsistent formatting: The figures are cited as Fig. in the main text. However, they are Figure in the caption.

The following sentence needs clarification. “The best binding-free energy of ligand-PDE5 complex files were chosen for MD simulation investigations”. What is the best binding-free energy? Is it the protein-ligand conformation with the smallest dG score?

If I understand the docking-simulation workflow correctly, only the bound structure with the highest affinity (smallest docking score) is simulated. However, in most cases, the docking scores of the top-n binding poses given by docking programs such as Autodock Vina are extremely similar. For instance, the top-2 or top-3 binding poses could have similar scores with differences smaller than 0.1, which suggests that the scoring function used in the docking program already has trouble differentiating the relative stability of these bound structures. However, the structural RMSDs could be very huge (e.g., ~ 2 Å), which indicates these docked poses are significantly different. This technically poses a problem in the selection of docking results for later simulation investigation, especially when the simulation is performed in an unbiased way. If the validity of this binding pose selection procedure is not checked, it is hard to say whether the simulation outcome is biased or not. Therefore, I strongly recommend the authors to run their simulations for protein-ligand complex initiated from top-2 or top-3 bound structures given by docking programs.

The parameters used for proteins and ligands are not mentioned.

In Section 2.7, the authors write that a cubic periodic box of 0.5 nm is constructed and the protein-edge distance is larger than 0.1 nm. However, such a setting is unacceptable in molecular simulations. Finite-size artifacts would be very significant and the simulation outcome is extremely biased.

Wrong abbreviation: MMPBSA -> MM/PBSA

The simulation tool is employed to obtain ensemble averages of various observables. However, no statistical error is reported for statistical quantities in many tables such as Table 6.

The backbone RMSDs reported in Table 6 and Figure 10 are very huge, which suggests that the energetically favorable region sampled in simulations is significantly different from the reference frame (often experimental structure). There could be several reasons triggering this phenomenon, e.g., inaccurate force field, improper modelling condition (temperature, ion concentration), and significant difference between the experimental condition and the simulated ensemble. The authors should investigate this part more thoroughly to really provide an answer, instead of simply presenting an observation.

The length of the manuscript seems too long compared with the analyses done in this work. The absence of supporting information could be the reason. Many Figures and Tables should be moved to the supporting information.

Author Response

Reviewer 3

Comments on molecules-1735970:

 The current paper reports a combined experimental and computational screening of bioactive compounds from Mimosa pudica L. The protein-binding performance of four top-binding compounds and their toxicity behavior were checked further to validate their potential use. Although many findings are biologically relevant and could be helpful to the molecules audience, significant improvements are required in many parts of the paper, especially the simulation part. Although I would like to consider a revised version for publication, they need to re-do the whole simulation part carefully. Please note that the reviewers’ comments are supportive and are provided to improve the quality of the research rather than criticizing anything purposedly, and the authors should take these suggestions carefully in the intended spirit.

Many abbreviations defined in the manuscript were not mentioned later. For example, ED defined in the first sentence and SPC defined in Section 2.7 are never mentioned later. In that case, these abbreviations should not be defined at all. The authors should check this issue in the whole manuscript carefully.

 Answer: Now corrected in our revised manuscript.

Inconsistent formatting: The figures are cited as Fig. in the main text. However, they are Figure in the caption.

 Answer: Thank you for viewpoints, now all the mistakes are corrected in our revised manuscript, as per the suggestion recieved.

The following sentence needs clarification. “The best binding-free energy of ligand-PDE5 complex files were chosen for MD simulation investigations”. What is the best binding-free energy? Is it the protein-ligand conformation with the smallest dG score?

 Answer: That is typo error – it should be best binding affinity. Now corrected in our revised manuscript

If I understand the docking-simulation workflow correctly, only the bound structure with the highest affinity (smallest docking score) is simulated. However, in most cases, the docking scores of the top-n binding poses given by docking programs such as Autodock Vina are extremely similar. For instance, the top-2 or top-3 binding poses could have similar scores with differences smaller than 0.1, which suggests that the scoring function used in the docking program already has trouble differentiating the relative stability of these bound structures. However, the structural RMSDs could be very huge (e.g., ~ 2 Å), which indicates these docked poses are significantly different. This technically poses a problem in the selection of docking results for later simulation investigation, especially when the simulation is performed in an unbiased way. If the validity of this binding pose selection procedure is not checked, it is hard to say whether the simulation outcome is biased or not. Therefore, I strongly recommend the authors to run their simulations for protein-ligand complex initiated from top-2 or top-3 bound structures given by docking programs.

Answer: We appreciate the reviewer for their critical review of our article. We accept Autodock vina algorithm generates many poses during the docking. When we analyses the result, it will produce three data such as Binding affinity, RMSD LB and RMSD UB, in which the top pose 1 will have high binding affinity and 0 for RMSD LB and RMSD UB and for the other poses there is differences in RMSD LB and RMSD UB values. The RMSD values get changed during the docking, hence that pose won’t be stable during dynamic simulation. Hence in this regard we performed dynamic simulation for the top pose of protein-ligand complexes.

The parameters used for proteins and ligands are not mentioned.

 Answer: All the parameters included in our revised article.

In Section 2.7, the authors write that a cubic periodic box of 0.5 nm is constructed and the protein-edge distance is larger than 0.1 nm. However, such a setting is unacceptable in molecular simulations. Finite-size artifacts would be very significant and the simulation outcome is extremely biased.

Answer:  While carrying out the system building before running dynamics, we have cross verified the box size, we found the protein was packed properly inside the box, hence we fixed cubic periodic box to 0.5 nm.

Wrong abbreviation: MMPBSA -> MM/PBSA

Answer: Now corrected in our revised manuscript.

The simulation tool is employed to obtain ensemble averages of various observables. However, no statistical error is reported for statistical quantities in many tables such as Table 6.

Answer: In table 6, we try to report the average of RMSD and RMSF for 5000 frames, hence we haven’t applied any statistical tool.

 The backbone RMSDs reported in Table 6 and Figure 10 are very huge, which suggests that the energetically favorable region sampled in simulations is significantly different from the reference frame (often experimental structure). There could be several reasons triggering this phenomenon, e.g., inaccurate force field, improper modelling condition (temperature, ion concentration), and significant difference between the experimental condition and the simulated ensemble. The authors should investigate this part more thoroughly to really provide an answer, instead of simply presenting an observation.

 Answer: We thank the reviewer for giving a critical point to analyses and fine tune our research study. As per the suggestion given by the reviewer, we made a small trial run of 10 ns with a proper force field and ensemble condition, for all the complex and apoprotein, we found same sort of result pattern. As we went through many literatures in finding the reason for the question raised by the reviewer, we found the reduction in RMSD of complex in comparison with apo, may due to the formation of strong hydrogen bonding interaction on the active pockets of protein.

The length of the manuscript seems too long compared with the analyses done in this work. The absence of supporting information could be the reason. Many Figures and Tables should be moved to the supporting information.

Answer: Thank you sir, as per your suggestion, the Figures 10-14 are combined in one single image as mentioned as Fig 10.

Round 2

Reviewer 1 Report

The authors are addressed all of my comments and the manuscript is acceptable

Author Response

The authors appreciate reviewer 2 for their critical review and thanks for giving valuable suggestions.

Reviewer 2 Report

Authors referred to all my comments.

Reviewer 3 Report

Comments on molecules-1735970.R1:

The authors endeavored to do some improvements in this revised manuscript. However, several points about scientific rigor raised in my previous comments are still not addressed.

Statistical observables reported in Table 6 do not have statistical uncertainty. This is unacceptable in modern molecular simulations.

Compared with the computational effort and analysis work done in this paper, the manuscript is too long. This greatly hinders the readability of the manuscript and should be improved. For example, statistics in Table 3 and plots in Figure 10 are of little importance and could be moved to SI.

My comments on re-running simulations with top-2 or top-3 binding pose to check the impact of the initial condition (docked pose) are not responded reasonably. Initial condition could have a significant impact on the simulation outcome, including the interaction network. If this check is not properly performed, the quality of the conformational search is not guaranteed and the computational results are questionable.

The simulation setup that I commented on, a cubic periodic box of 0.5 nm and the protein-edge distance larger than 0.1 nm, does not seem reasonable.

Author Response

The authors endeavored to do some improvements in this revised manuscript. However, several points about scientific rigor raised in my previous comments are still not addressed.

Statistical observables reported in Table 6 do not have statistical uncertainty. This is unacceptable in modern molecular simulations.

Answer: As per the suggestion of reviewer, the statistical value (mean±SD) is incorporated in Table 6, in the revised manuscript. 

Compared with the computational effort and analysis work done in this paper, the manuscript is too long. This greatly hinders the readability of the manuscript and should be improved. For example, statistics in Table 3 and plots in Figure 10 are of little importance and could be moved to SI.

Answer: Thanks for your valuable comments. Table 3 is now moved into supplementary file, and Figures 3 to 7 are merged into a single image, as Figure 3. Moreover, molecular dynamics simulation images are very essential for our research work, so we already combined in a single image, now as Figure 6.  

My comments on re-running simulations with top-2 or top-3 binding pose to check the impact of the initial condition (docked pose) are not responded reasonably. Initial condition could have a significant impact on the simulation outcome, including the interaction network. If this check is not properly performed, the quality of the conformational search is not guaranteed and the computational results are questionable.

Answer: We thank the reviewer for raising this concern. The idea behind choosing the protein-ligand complex with the lowest binding energy is the lowest binding energy elucidates that the complex has exhibited highest binding affinity. In other words, it deals with the strongest bonding when compared to the other poses. We agree with the reviewer's statement on the importance validation of binding poses but this will not be acceptable for this study. In this study, we have tried to screen multiple compounds and identify the best compound based on the binding affinity. If we had to select the single compound's binding based on a pose, then we might end up in an unbalanced study. For example, compound-1 might exhibit the best conformer in pose-2, compound-2 might exhibit the best conformer in pose-3, and compound-3 might exhibit the best conformer in pose-1, and so on. In this scenario, we don't have a standard comparison except the scale as RMSD=0. This might lead to ambiguity in the study. This is the major reason for choosing only the best pose at RMSD=0. However, we value the suggestion of the reviewer and we will implement his/her suggestion in our future studies. Once again, we thank the reviewer for the valuable time and suggestion.

The simulation setup that I commented on, a cubic periodic box of 0.5 nm and the protein-edge distance larger than 0.1 nm, does not seem reasonable.

Answer: The authors appreciate the reviewer’s for their critical review, after the cross verification on the script, we have corrected the box size in our revised manuscript.